# Association Between Psychobehavioral Factors and the Increased Eating Rate of Ultra-Processed Versus Non-Ultra-Processed Meals in Individuals with Obesity: A Secondary Analysis of a Randomized Trial

**DOI:** 10.3390/nu17132236

**Published:** 2025-07-05

**Authors:** Ludmila de Melo Barros, Vanessa Amorim Peixoto, Guilherme César Oliveira de Carvalho, Micnéias Róberth Pereira, Rodrigo Tenório Lins Carnaúba, Nassib Bezerra Bueno

**Affiliations:** Laboratório de Nutrição e Metabolismo (LANUM), Programa de Pós-Graduação em Nutrição (PPGNUT), Faculdade de Nutrição, Universidade Federal de Alagoas, Campus AC Simões, Av. Lourival Melo Mota, s/n, Cidade Universitária, Maceió 57072-900, AL, Brazil; ludmila.barros@fanut.ufal.br (L.d.M.B.); vanessa.peixoto@fanut.ufal.br (V.A.P.); guilherme.carvalho@fanut.ufal.br (G.C.O.d.C.); micneias.pereira@fanut.ufal.br (M.R.P.); rodrigo.carnauba@fanut.ufal.br (R.T.L.C.)

**Keywords:** food intake, feeding behavior, eating disorders, body image

## Abstract

**Background/Objectives**: A faster eating rate is associated with increased energy intake and risk of obesity. High consumption of ultra-processed foods (UPFs) is associated with a faster eating rate. Psychobehavioral aspects, such as body image self-perception, eating disorders, and anxiety, may modulate this eating behavior. Therefore, this study examined the moderating role of psychobehavioral factors in the association between meal type (UPF vs non-UPF) and eating rate among individuals with obesity. **Methods**: It is a secondary analysis of a randomized, parallel clinical trial conducted with 39 adults who have obesity. Participants were assigned to consume either a UPF-only composed meal or a UPF-free meal, both of which were isoenergetic (~550 kcal). Psychobehavioral variables (food addiction—mYFAS 2.0, body image perception and satisfaction—Silhouette Rating Scale, eating disorders—EAT-26, and anxiety—GAD-7) were assessed. Eating rate was measured in kcal/min. **Results**: Body image perception and satisfaction significantly interacted with the type of meal. In the UPF group, lower body image dissatisfaction was associated with a higher eating rate (β = 4.79 kcal/min; 95% CI: 1.40; 8.19; *p* = 0.007), while a higher body image perception score was associated with a lower eating rate (β = −4.61 kcal/min; 95% CI: −8.57; −0.65; *p* = 0.024). No significant associations were observed for food addiction scores, eating disorders or anxiety. **Conclusions**: Body image modulates the eating rate in the context of UPF consumption. These findings suggest that interventions against obesity should consider individual psychobehavioral characteristics.

## 1. Introduction

Obesity imposes an estimated economic burden of USD 2 trillion annually [1]. Globally, approximately 890 million adults have excess body fat that adversely affects their quality of life [2] and is associated with increased risks of morbidity and all-cause mortality [3,4]. Despite advances in identifying the factors underlying the energy imbalance that contribute to its etiology, the limited effectiveness of current obesity treatments suggests an incomplete understanding of the mechanisms driving excessive food intake [5,6].

The eating rate is a key aspect of food microstructure that influences energy intake [7]. A systematic review with meta-analysis of 22 clinical trials demonstrated that faster eating leads to higher energy intake, with the effect size sensitive to the magnitude of the change in eating rate [8]. Shorter oral processing time and limited chewing activity also tend to increase post-meal hunger and desire to eat [9], factors positively associated with excess body weight [10,11]. Furthermore, individuals with obesity exhibit faster eating rates, correlated with larger bite sizes, shorter oral exposure times per bite, and higher energy intake, suggesting a typically obesogenic eating style [12,13].

This pattern of behavior is influenced by the rising availability of ultra-processed foods (UPFs) in the food system [14]. These industrialized formulations, which are generally composed of food additives to enhance sensory characteristics and are even absent from whole foods [15], undergo degradation of the food matrix, resulting in a softer texture that is easier to chew [16,17]. As a result, UPF consumption exhibits a significantly higher eating rate compared with other processing categories [18], even in individuals with obesity [19,20]. This may partially explain the sustained increases in energy intake and body weight attributed to the consumption of these products [18,20,21].

In addition to environmental influences, psychobehavioral factors also explain individual differences in susceptibility to overeating [22]. For example, compulsive eating disorders impose a higher risk of losing control over appetite [23], leading to higher energy intake and lower satiety in overweight individuals [24]. Patients with bulimia nervosa also eat twice as fast as those without the condition [25]. Meanwhile, generalized anxiety disorder has been positively associated with hunger and disinhibition behaviors, which lead to overeating [26]. In the specific context of UPF consumption, body image perception and dissatisfaction are psychosocial drivers that promote greater intake of these products [27]. In turn, food addiction, a phenomenon recognized for its similarities to substance use disorders, also has an impact on the consumption of UPF [28,29].

Hence, it is essential to deepen our understanding of the synergistic effect between individual and environmental determinants in disruptive eating behavior [30]. Therefore, the present study aims to identify psychobehavioral factors that interact with the increased UPF eating rate, previously identified in a randomized clinical trial with individuals with obesity [19]. This secondary analysis offers insights into more integrated approaches to treating overeating and obesity [31].

## 2. Materials and Methods

### 2.1. Experimental Design

This is a secondary analysis of a randomized, parallel clinical trial with two research arms [19]. Due to the nature of the intervention, it was not possible to conduct a double-blind study. The present study is reported in accordance with the Consolidated Standards of Reporting Trials (CONSORT) guidelines [32]. The trial was approved by the Research Ethics Committee of the Federal University of Alagoas (UFAL) (CAAE registration number: 69062123.2.0000.5013) and submitted to the Registry of Brazilian Clinical Trials (RBR-56nsh92). Eligible candidates who satisfied all inclusion criteria provided written informed consent prior to enrollment, thereby formalizing their study participation.

### 2.2. Location, Sample, and Sampling

The investigation was carried out at Nutrition and Metabolism Laboratory (LANUM), affiliated with the Faculty of Nutrition (FANUT) at the Federal University of Alagoas (UFAL). The non-probabilistic convenience sampling method was used between July and August 2023. Recruitment strategies included announcements on campus and social media accounts of the Laboratory and UFAL, as well as the university’s official website. The study enrolled adult participants aged 19–60 years with obesity, requiring fulfillment of at least two diagnostic criteria: (1) BMI ranging from 25.0 to 40.0 kg/m^2^; (2) gender-specific abdominal circumference thresholds (≥88 cm for female participants; ≥102 cm for males); or (3) elevated adiposity levels (≥35% body fat for women; ≥25% for men) as measured by bioelectrical impedance analysis [33]. Participants were not included if they were taking chronic medications (e.g., antibiotics, glucose-lowering agents, antiretrovirals, immunomodulators, or psychotropic drugs), had food hypersensitivities or dietary restrictions that could interfere with test meal consumption, presented any clinical conditions preventing anthropometric or metabolic assessments, were menopausal, pregnant, or lactating women, or had undergone bariatric procedures. Additionally, individuals who consumed less than 95% of the test meal were excluded.

### 2.3. Randomization and Allocation

A computerized randomization scheme using R statistical software (v4.3.1) was implemented. The algorithm generated 100 pseudorandom values (0–1 range), subsequently rounded to integers (0 = control; 1 = UPF group). To ensure allocation concealment, the randomization sequence was securely maintained by an independent researcher not involved in participant screening or enrollment. Group assignments were only determined after completion of all baseline measurements, with the allocation information subsequently revealed by a blinded study team member prior to each participant’s test meal session.

### 2.4. Procedures

On the scheduled testing day, participants reported to the research center between 7:00 and 9:00 AM following a 12 h overnight fast and 24 h abstinence from vigorous physical activity, caffeine, and energy drinks. Upon arrival, researchers performed baseline evaluations including resting energy expenditure measurements and heart rate variability assessments, along with other biological parameter analyses. Between 8:00 and 10:00 AM, participants were escorted to a private, climate-controlled (22–24 °C) testing room where they received their assigned test meal (breakfast) with instructions to consume it completely. Additional methodological details are available in Galdino-Silva et al. [19]

### 2.5. Test Meals

Two isocaloric test meals matched for macronutrient composition, fiber content, sodium levels, and energy density (1.4 kcal/g), differing solely in their degree of food processing, were compared. Both meals provided approximately 550 kcal, containing 70 g carbohydrates, 22 g lipids, 18 g protein, 9 g dietary fiber, and 1300 mg sodium (Table 1). The ultra-processed food (UPF) condition featured exclusively industrial formulations: packaged toast with processed cheese and ham, commercial strawberry jam, margarine, and fiber-fortified guava juice. In contrast, the control meal comprised only fresh or minimally processed ingredients: artisanal bread with extra-virgin olive oil, pan-fried chicken eggs (using soybean oil and salt), and freshly prepared guava juice sweetened with honey (Figure 1). All meals were prepared under controlled conditions in the Technical and Dietetics Laboratory of FANUT/UFAL one hour before serving, following standardized food safety protocols for research-grade meal preparation.

### 2.6. Food Addiction

Food addiction was measured using a modified Yale Food Addiction Scale 2.0 (mYFAS 2.0) applied at baseline [34]. This scale was translated and cross-culturally validated into Brazilian Portuguese, demonstrating internal consistency and an adequate factor structure [35,36]. It consists of 13 questions, of which 11 represent symptoms related to the individual’s eating behavior that refer to aspects of substance use disorders in the *Diagnostic and Statistical Manual of Mental Disorders, Fifth Edition* (DSM-5) [37], and two refer to clinical impairment or distress. Each question is answered according to the frequency with which the symptom occurs, ranging from “never” to “every day,” and there is a threshold for each of them so that the criteria for the symptom are met. Finally, the 11 symptoms were summed up to create a symptom score. Individuals who presented two or more symptoms and met the thresholds for any of the clinical impairment or distress questions were classified as having food addiction [34].

### 2.7. Body Image Self-Perception

To assess body image self-perception, the Silhouette Rating Scale, a validated and adapted instrument for the Brazilian population [38], was employed. The scale consists of 15 white silhouette figures on a black background, with corresponding BMI averages ranging from 12.5 kg/m^2^ at the first silhouette to 47.5 kg/m^2^ at the last, with an interval of 2.5 kg/m^2^ between each figure. At baseline, participants were presented with 15 silhouette figures in an ascending orderly series and asked the following questions: “Which figure best represents your current size?”, “Which figure would you consider to be the ideal size for your gender in general?” and “Which figure represents the size you would like to have?”. The discrepancy between the ideal self-image and the current self-image at the time of the questionnaire was calculated by subtracting the numbers of the chosen silhouettes. Thus, the closer the subtraction result is to zero, the lower the participant’s dissatisfaction with body image. Furthermore, the ability to estimate the self-perception body image was also verified by comparing the self-image chosen as “current” and the individual’s BMI.

### 2.8. Generalized Anxiety Disorder Scale (GAD-7)

The GAD-7 scale, translated and validated for Brazilian Portuguese, was applied at baseline to identify potential cases of generalized anxiety disorder [39]. It consists of seven items, arranged on a four-point Likert scale ranging from 0 to 3, where 0 represents “not at all” and 3 “almost every day”; the score is summed and varies from 0 to 21 points. A probable case of generalized anxiety disorder was considered when the score obtained was equal to or greater than 10 [40].

### 2.9. Eating Attitudes

To assess eating behaviors and attitudes that may be associated with eating disorders, the Eating Attitudes Test (EAT-26) [41] was applied at baseline. The EAT-26 was translated and validated into Portuguese [42] and consists of 26 items, which are divided into three factors: “diet”, “bulimia and preoccupation with food”, and “oral control”. The questions are arranged on a Likert-type scale, which varies between “always = 3 points”, “often = 2 points”, “sometimes = 1 point”, “rarely = 0 points”, “almost never = 0 points”, and “never = 0 points”. After summing up the scores assigned to each item, individuals who obtained a score higher than 21 [41,42] were considered at risk for eating disorders.

### 2.10. Eating Rate

The entire meal consumption process was video-recorded for subsequent analysis of eating rate parameters. Participants were informed about the recording procedure exclusively after meal completion. Meal duration was operationally defined as the interval between the first bite and final swallow. Based on these recordings, calories consumed per minute (kcal/min) were calculated for both solid and liquid components using established methodologies [18,43].

### 2.11. Additional Data

Demographic, socioeconomic, clinical, and lifestyle data, including sex, age, educational attainment, alcohol and tobacco use patterns, chronic medication regimens, pre-existing medical conditions, and socioeconomic status (classified according to Brazilian Economic Classification Criteria [44]), were obtained through a validated research questionnaire. Anthropometric measurements included the following: (1) body weight, assessed to the nearest 100 g using a calibrated portable digital scale (capacity: 150 kg) with participants wearing light clothing and no footwear; (2) standing height, measured with 1 mm precision using a portable stadiometer (range: 0–2.2 m); (3) waist circumference, determined at the midpoint between the lowest rib and iliac crest using a non-stretch tape measure with participants in standardized anatomical position; and (4) body composition parameters (fat mass percentage, fat-free mass, and total body water) evaluated via tetrapolar bioelectrical impedance analysis (RJL Quantum IV, RJL Systems). Subjective appetite sensations were quantified after 12 h fasting state using a 100 mm visual analog scale (VAS) assessing four domains: hunger (“How hungry do you feel now?”), satiety (“How full do you feel right now?”), desire to eat (“How much do you want to eat now?”), and capacity to eat (“How much do you think you can eat now?”), with anchors ranging from “nothing/none” (0 mm) to “excessive” (100 mm) [18,45].

### 2.12. Sample Size Calculation

Estimating a standardized effect size (Cohen’s d) of 0.9, indicating a “large” effect size, considering the post-prandial measure of hunger as the primary outcome, with a statistical power of 80% and a significance level of 5%, 21 patients per group were required to achieve significant results. Although studies investigating effects on eating rate may vary greatly in sample size [46], other clinical trials have similar sample sizes [47,48].

### 2.13. Statistical Analysis

Multivariable linear regression was performed to evaluate the effect of psychobehavioral factors (the symptom score of FA, EAT-26, GAD-7, as well as body perception and satisfaction), used as independent variables in the different models, and the eating rate, used as outcome in various models. Initially, we conducted separate analyses by group, and thereafter, we ran a complete model, calculating the interaction term between the independent variable and the intervention group (UPF meal or control meal). All analyses were adjusted for age and sex. The results were reported as beta coefficients, along with their corresponding 95% confidence intervals and *p*-values. Pearson’s correlation coefficient was calculated between body perception and satisfaction to explore the relationship between complementary aspects of body image.

Continuous variables were expressed as mean (standard deviation), while categorical variables were reported as frequency counts and percentages. A *t*-test and Fisher’s exact test were also used to compare means between groups. All analyses were conducted using JAMOVI software, version 2.3.28 (Sidney, Australia), with an alpha value of 0.05 for statistical significance.

## 3. Results

Of the 94 individuals assessed for eligibility, 49 were excluded (see Figure 2 for details), resulting in 43 participants with obesity being allocated to the intervention or control group. However, four participants were excluded because they failed to consume at least 95% of the test meal (Figure 2). Consequently, the final sample consisted of 39 individuals (control group, n = 20; UPF group, n = 19), predominantly female, who self-identified as Black, non-alcoholic, sedentary, and with a high school education level. None of the participants were smokers, had diagnosed noncommunicable chronic diseases, or exhibited signs of fever and tachycardia. Baseline characteristics are shown in Table 2, with no statistically significant differences between groups.

Table 3 presents the results of the linear regression analysis examining the associations between psychobehavioral variables and intervention groups, including interaction and subgroup analyses. A significant interaction was found between body image satisfaction and meal type on eating rate (β = 4.79 kcal/min; 95% CI: 1.40; 8.19; *p*-value×group = 0.007), in contrast to the model without adjustment for sex and age (β = 2.80 kcal/min; 95% CI: −2.07; 7.68; *p*-value×group = 0.251). Specifically, in the UPF group, higher body image satisfaction, reflected by scores closer to zero, was associated with an increased eating rate, suggesting that individuals with less dissatisfaction with their bodies consumed food more quickly when exposed to UPF meals (Figure 3), which contrasted with the control group.

Regarding body image perception, the results also revealed a significant interaction with meal type, with (β = −4.61 kcal/min; 95% CI: −8.57; −0.65; *p*-value×group = 0.024) and without adjustment (β = −6.11 kcal/min; 95% CI: −11.37; −0.85; *p*-value×group = 0.024). In comparison with the control group, the UPF group’s higher body image perception scores were associated with a decreased eating rate, as shown in Figure 4. A significant negative association was also revealed between body image perception and satisfaction (r = −0.522, *p* < 0.001), indicating that as perception scores increase, satisfaction tends to decrease (Figure A1).

No significant associations were found for food addiction symptoms, EAT-26, or GAD-7 scores in either group or the overall sample (Table 3).

## 4. Discussion

This study’s results suggest that body image satisfaction and perception affect the eating rate according to the meal’s processing level. In the UPF meal group, decreased body image dissatisfaction was associated with a faster eating rate, whereas in the control meal group, it was associated with a slower eating rate. In addition, participants in the UPF meal group with higher body image perception values tended to eat more slowly, in contrast to the increased eating rate in the control meal group. The eating rate did not show a significant association with the screening of risk behaviors for eating disorders, anxiety levels, and food addiction symptom scores, which highlights the specific influence of body image on this eating behavior in the present study.

Although clinical trials have consistently reported faster eating rates with UPF meals due to their physical properties, such as soft texture and high energy density, which encourage larger bites, less chewing, and shorter meal duration [18,20,49,50], our results suggest that psychobehavioral variables, particularly body image satisfaction and perception may influence counteract this trend. The contrast with the pattern observed by Galdino-Silva et al. (2024) [19], who reported significantly faster eating rates among individuals with obesity consuming the same UPF meals, suggests that the effect of UPFs is not uniform across populations. Instead, it may be modulated by individual psychological factors, which influence how people engage with these foods.

Body image is a multidimensional psychological construct that reflects how individuals perceive, think, and feel about their own bodies. Its perceptual component refers to the accuracy with which this perception aligns with actual body size [51]. In our study, higher scores of self-body perception, either as a smaller discrepancy between actual and perceived BMI or as an overestimation of body size, were associated with a reduced eating rate during the consumption of the UPF meal. The attitudinal component of self-body image, which encompasses emotional and cognitive responses to one’s own body, such as satisfaction or dissatisfaction with appearance, also plays a role in eating behavior. These attitudes often translate into efforts to alter body weight, typically shaped by internalized ideals of appearance [52,53]. Interestingly, our findings indicate that higher levels of body image satisfaction, even when still within the negative range, were associated with increased eating rates during UPF consumption. Given the established connection between self-image and emotional responses [51], individuals who identify with a higher body weight tend to show greater body image perception scores and lower levels of satisfaction, reinforcing the idea that how one perceives and evaluates their body can shape patterns of eating behavior, particularly eating rate.

It is known that greater dissatisfaction with body image tends to lead to maladaptive eating patterns, such as increased compensatory restriction [54]. Such behavior involves the intentional and continuous limitation of food intake, often guided by self-imposed dietary rules that restrict the consumption of foods considered bad [55]. This stigmatization of “good” or “bad” foods may be rooted in the adverse health outcomes linked to the consumption of UPF [56,57], leading to discouragement of its consumption at the public health level, and thus contributing to the stigmatization of these products [15,58].

Given this context, our finding that individuals in the UPF group who reported greater body image dissatisfaction due to the desire to be smaller than their perceived body size tended to eat more slowly becomes particularly noteworthy. This pattern suggests that when confronted with a stigmatized food category like UPF, individuals with heightened body image concerns adopt a slower eating rate, either consciously or unconsciously, as a form of self-regulation. Such behavior may reflect an internal conflict between the appeal of consuming the food and concerns about its perceived negative impact on their body image [54,59]. These observations underscore the need to explore further the synergistic effect between UPF consumption and body image, in order to clarify the mechanisms underlying this relationship that may explain the reduction in eating rate.

The relationship between UPF consumption and psychobehavioral factors has already been explored in the literature. In a systematic review, Pereira et al. [60] investigated the associations between UPF consumption and eating disorders, food addiction, and body image dissatisfaction. Their findings revealed a significant association between food addiction, bulimia, binge eating, and unspecified eating disorders with UPF consumption. Considering that eating rate tends to be higher when consuming UPF, these psychological traits might also be associated with the eating rate. However, our findings did not show any significant interaction between UPF consumption and the risk of eating disorders in terms of eating rate. This may be partially explained by the limitations of the assessment tool used (EAT-26), which predominantly targets behaviors linked to anorexia nervosa and bulimia nervosa, disorders less prevalent among individuals with overweight and obesity compared with binge eating disorder [61,62]. Moreover, the similarity between the intervention and control meals in terms of texture, macronutrient composition, and energy density, factors that influence eating rate [63], may have attenuated the potential effects related to food addiction, which is more likely to emerge with the consumption of hyperpalatable and energy-dense items [64].

It is worth noting that body perception among individuals with excess weight is frequently distorted, often involving underestimation or overestimation of body size [65,66,67]. A systematic review and meta-analysis reported that individuals with obesity exhibit higher levels of body dissatisfaction compared with those with an adequate weight [68]. This dissatisfaction tends to be more pronounced among women, as consistently documented in the literature [69,70], underscoring the importance of considering gender in our analytical model. However, in our sample, body perception presented a more uniform pattern between male and female participants.

Some limitations should be acknowledged when interpreting these results. This is a secondary analysis of a randomized clinical trial, which means that causal inference cannot be drawn because the hypotheses tested here were not the primary objective of the original design. In addition, sample losses resulting from the exclusion of participants who did not consume the entire meal offered reduced statistical power. Although video recording may have influenced participants’ eating behavior due to the tendency to alter behavior when being observed, it remains the least intrusive method for capturing eating episodes [71,72,73]. About body image assessment, comparisons with the literature are limited by the diversity of tools used to measure this construct [68]. Despite this, body silhouette scales demonstrate predictive validity by using standardized images that reduce the diversity of individual abstractions [74,75]. In particular, the scale proposed by Kakeshita et al. [38] was developed for the Brazilian population, reflecting its particular cultural and socioeconomic context [38]. In this sense, the findings provide relevant evidence on the potential synergy between the environmental impact of UPF consumption and individual psychological traits. Given the limited effectiveness of traditional information-based campaigns in improving dietary habits, acknowledging the interaction between body image and UPF intake may enable more strategic allocation of public health resources and enhance the impact of interventions targeting obesity. These insights highlight the importance of tailoring policy strategies to the specific needs of psychologically vulnerable subgroups [30,31,76].

## 5. Conclusions

This study revealed a significant interaction between body image and the food processing level on the eating rate of individuals with obesity. Higher body perception scores, as well as higher levels of dissatisfaction related to the desire for a smaller body than perceived, were associated with a slower eating rate during the consumption of UPF, contrary to the pattern generally expected for this type of food. These findings suggest that psychological factors, particularly the relationship with one’s own body image, can modulate the behavioral effects commonly attributed to UPF consumption. This highlights the relevance of future research focusing on populations with body image distortions and dissatisfaction, in order to better understand the nuanced impacts of UPF consumption in these groups.

## Figures and Tables

**Figure 1 nutrients-17-02236-f001:**
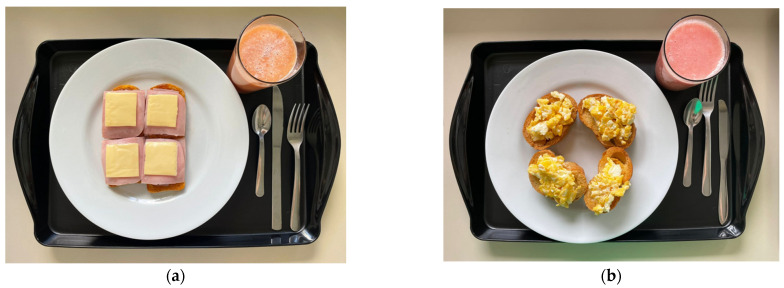
(**a**) Test meal rich in ultra-processed foods offered to the group of participants allocated to the UPF Meal group on the day of the clinical trial. (**b**) Test meal without ultra-processed foods offered to the group of participants allocated to the Control Meal group on the day of the clinical trial. Reproduced from Galdino-Silva et al. [19].

**Figure 2 nutrients-17-02236-f002:**
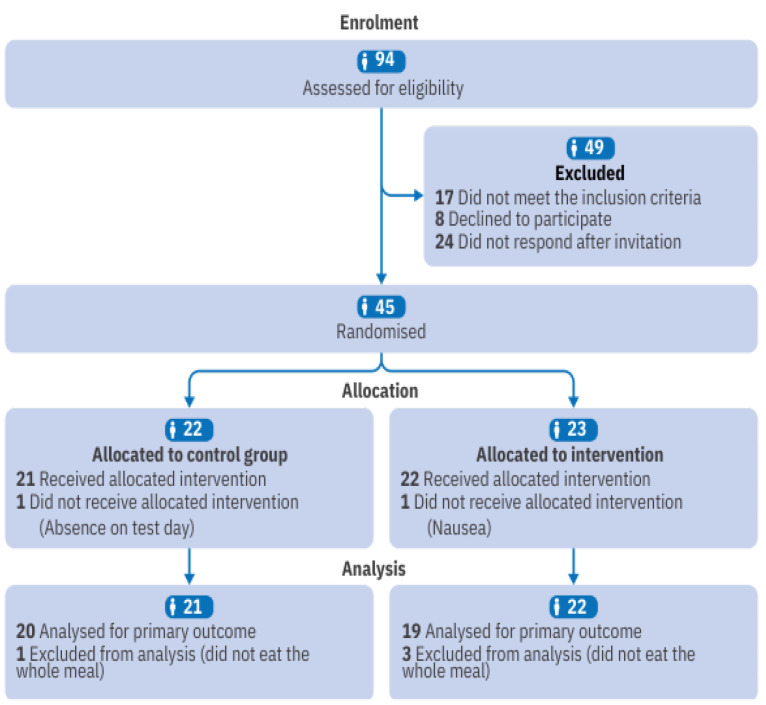
CONSORT 2025 flow diagram of participant progress through the trial.

**Figure 3 nutrients-17-02236-f003:**
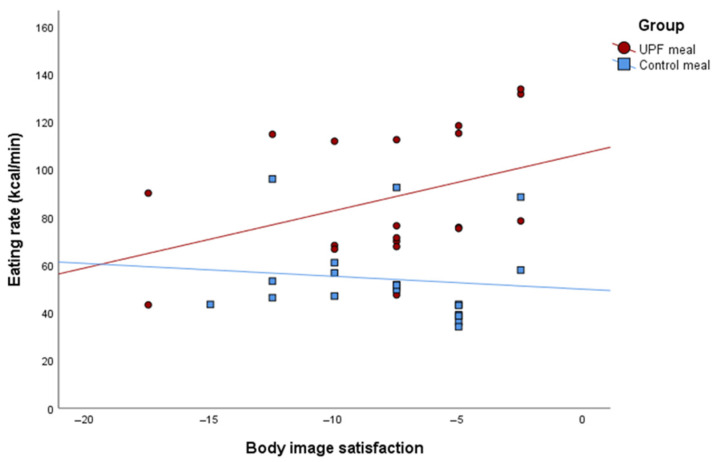
Interaction plot from the linear regression between body image satisfaction and eating rate (kcal/min) across groups, adjusted for sex and age.

**Figure 4 nutrients-17-02236-f004:**
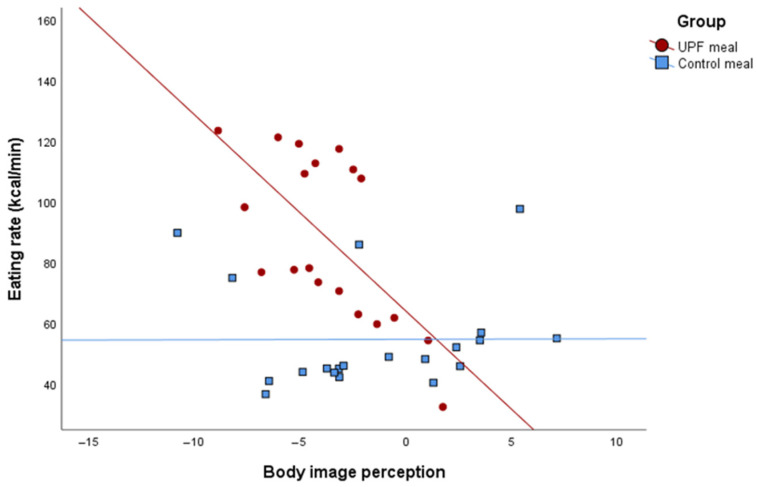
Interaction plot from the linear regression between body image perception and eating rate (kcal/min) across groups, adjusted for sex and age.

**Table 1 nutrients-17-02236-t001:** Composition of UPF and non-UPF meals in the randomized clinical trial.

UPF Meal					Control Meal				
Bauducco^®^ Traditional Toast (04 units)—30 g Polenghi^®^ dish cheese (03 fts)—54 g Perdigão^®^ Ham (2.5 fts)—50 g Purifruti^®^ Strawberry Jelly (02 tablespoons)—40 g Primor^®^ Margarine (01 dessert spoon)—15 g Maratá^®^ guava juice (01 glass)—200 mL Nestlé^®^ FiberMais fiber module (1 tablespoon)—9 g					Regular bakery bread toast (01 unit)—50 g Chicken egg (02 units)—100 g Soybean vegetable oil (01 teaspoon)—5 mL Refined salt (01 coffee spoon)—2 g Extra virgin olive oil (01 tablespoon)—8 mL Guava pulp juice (1 glass)—200 mL Bee honey (01 tablespoon)—20 g				
Meal	KCAL	PTN (g)	PTN (%)	CHO (g)	CHO (%)	LIP (g)	LIP (%)	Fibers (g)	Na (mg)
UPF	557.3	18.4	13.2	71.3	51.2	21.9	35.4	8.5	1493.3
No UPF	559.7	18.7	13.4	70.6	50.5	22.1	35.5	8.5	1267.5

UPF: ultra-processed food; g: grams; fts: slices; mL: milliliters; KCAL: kilocalories; PTN: proteins; CHO: carbohydrates; LIP: lipids; Na: sodium; mg: milligrams; %: percentage. Reproduced from Galdino et al. [19].

**Table 2 nutrients-17-02236-t002:** Baseline descriptive characteristics of the study sample.

Variables	Control Group (n = 20)		UPF Group (n = 19)		*p*-Value ^a^
	n	%	n	%	
Sex					0.17
Female	16	80.0	11	57.9	
Male	4	20.0	8	42.1	
Skin color					0.79
White	5	25.0	6	31.6	
Brown	3	15.0	1	5.3	
Black	12	60.0	12	63.2	
Alcohol consumer					0.75
Yes	9	45.0	10	52.6	
No	11	55.0	9	47.4	
Physically inactive					0.75
Yes	11	55.0	9	47.4	
No	9	45.0	10	52.6	
Brazilian economic class system					0.29
A	1	5.0	1	5.3	
B1	1	5.0	0	0.0	
B2	4	20.0	6	31.6	
C1	7	35.0	6	31.6	
C2	7	35.0	3	15.8	
D/E	0	0.0	3	15.8	
Educational level					0.74
Completed high school	14	70.0	12	63.2	
Completed higher education	6	30.0	7	36.8	
	Mean	SD	Mean	SD	*p*-Value ^b^
Age (years)	29.9	7.91	28.0	6.57	0.42
Height (m)	1.64	0.10	1.68	0.09	0.16
Weight (kg)	82.09	15.41	91.39	17.10	0.08
BMI (kg/m^2^)	30.40	3.44	32.02	3.91	0.17
Waist circumference (cm)	92.90	12.86	98.15	10.63	0.17
Body fat percentage (%)	41.38	6.45	39.41	6.55	0.35
Fasting hunger	44.5	26.5	45.5	30.6	0.91
Fasting fullness	25.3	18.6	20.4	18.1	0.41
Fasting satisfaction	54.9	27.9	49.9	30.1	0.60
Fasting prospective food consumption	57.8	22.6	61.3	26.3	0.65

UPF: ultraprocessed food; SD: standard deviation; m: meters; kg: kilograms; kg/m^2^: kilograms per square meter; cm: centimeters; ^a^ *p*-Values obtained using Fisher’s exact test; ^b^ *p*-Values obtained using the independent samples *t*-test.

**Table 3 nutrients-17-02236-t003:** Association between psychobehavioral variables and eating rate (kcal/min) in a meal with or without UPF.

	Control Group (n = 20)			UPF Group (n = 19)			Overall Sample (n = 39)			
	Beta	95% CI	*p*-Value ^a^	Beta	95% CI	*p*-Value ^a^	Beta	95% CI	*p*-Value ^b^	*p*-Value × Group ^c^
Number of FA symptoms	1.68	−0.73; 4.10	0.15	0.85	−5.62; 7.32	0.78	1.70	−1.32; 4.72	0.26	0.89
EAT-26 score	0.30	−0.79; 1.41	0.56	−0.14	−2.18; 1.89	0.88	0.08	−1.01; 1.18	0.87	0.49
GAD-7 score	0.41	−0.77; 1.61	0.46	−0.50	−3.50; 2.49	0.72	0.29	−1.11; 1.71	0.67	0.23
Body satisfaction	−1.40	−3.20; 0.39	0.11	2.71	−0.14; 5.57	0.06	0.59	−1.17; 2.35	0.49	0.007
Body image perception	0.72	−0.56; 2.01	0.25	−3.10	−8.02; 1.80	0.19	0.01	−1.83; 1.85	0.98	0.024

UPF: ultra-processed food; 95% CI: 95% confidence interval; FA: food addiction; EAT-26: Eating Attitudes Test-26; GAD-7: Generalized Anxiety Disorder-7; ^a^ *p*-values refer to linear regression adjusted for sex, and age; ^b^ *p*-Values refer to linear regression adjusted for group, sex, and age; ^c^ *p*-Value for the interaction coefficient between group and each independent variables, adjusted by sex and age.

## Data Availability

The raw data supporting the conclusions of this article will be made available by the authors upon request.

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
