# Peer review of "Association Between Psychobehavioral Factors and the Increased Eating Rate of Ultra-Processed Versus Non-Ultra-Processed Meals in Individuals with Obesity: A Secondary Analysis of a Randomized Trial"

_nutrients, 2025, doi:10.3390/nu17132236_

Round 1
Reviewer 1 Report
Comments and Suggestions for Authors
I appreciate the opportunity to review this manuscript. The manuscript uses secondary data to analyze the relation between body satisfaction and eating rate. The manuscript uses well-documented instruments to measure body image and related aspects. My suggestions:
- The overall analysis is about the relation between body satisfaction and eating rate. Please keep it simple, when 3-4 variables are also included, the overall research becomes unnecessarily complicated.
- The sample size is relatively small, however, it is common for this type of studies. I would suggest to provide some examples to provide support on the sample size.
- I suggest to relate the findings to policy implications. How these findings can be useful?
Regards,
Author Response
- Comment 1: “The overall analysis is about the relation between body satisfaction and eating rate. Please keep it simple, when 3-4 variables are also included, the overall research becomes unnecessarily complicated.”
- Author’s response 1: Thank you very much for your observation. We understand the importance of maintaining clarity and focus in the analytical approach. However, given the exploratory nature of this secondary analysis, our aim was to investigate whether psychological variables could contribute to hypothesis generation in the novel context involving UPF in individuals with obesity. We believe even the absence of significant associations for the remaining psychological parameters still enriches the interpretation and supports future research directions, rather than complicating the analysis.
- Comment 2: “The sample size is relatively small, however, it is common for this type of studies. I would suggest to provide some examples to provide support on the sample size.”
- Author’s response 2: We thank the reviewer for highlighting this aspect. We have now included a sentence in the manuscript to address this point (lines 216-217): “Although studies investigating effects on eating rate may vary greatly in sample size [46], other clinical trials have similar sample sizes [47,48].”
- Comment 3: “I suggest to relate the findings to policy implications. How these findings can be useful?”
- Author’s response 3: We thank the reviewer for this valuable suggestion. In response, we have added the following sentence to the discussion to highlight the potential policy implications of our findings (lines 375-380): “Given the limited effectiveness of traditional information-based campaigns in improving dietary habits, acknowledging the interaction between body image and UPF intake may enable more strategic allocation of public health resources and enhance the impact of interventions targeting obesity. These insights highlight the importance of tailoring policy strategies to the specific needs of psychologically vulnerable subgroups [30,31,76].”
Reviewer 2 Report
Comments and Suggestions for Authors
The article presents a secondary analysis of a randomized clinical trial investigating the interaction between psychobehavioral factors and eating rate of ultra-processed (UPF) versus non-UPF meals in individuals with obesity. The methodology is rigorous, employing validated psychometric tools (mYFAS 2.0, EAT-26, GAD-7, Silhouette Scale), controlled experimental conditions, and multivariate regression analyses. The test meals were isoenergetic and comparable in composition, differing only in processing level, allowing for a controlled assessment of UPF effects.
The main findings indicate that body image perception and satisfaction modulate eating rate specifically in the UPF group, with no significant associations observed for food addiction, eating disorders, or anxiety scores. The analysis appropriately adjusts for age and sex, and significant interactions are statistically supported. The sample size (n=39) is small, which limits generalizability and statistical power. Furthermore, the study’s exploratory nature, as a secondary analysis, necessitates caution in causal interpretation.
The use of body image scales based on silhouette figures, although validated for the Brazilian population, presents limited comparability with international literature. Additionally, the choice of EAT-26, a tool more suited for detecting anorexia and bulimia nervosa, may have underestimated relevant eating pathology in an obese sample, particularly binge eating disorder.
To improve the manuscript:
-
Clarify the rationale for including specific psychometric instruments and acknowledge limitations of EAT-26 in obese populations.
-
Expand on the theoretical basis for the proposed mechanisms linking body image with eating rate.
-
Provide power calculations adjusted for the interaction models used.
-
Address the potential effect of video observation on eating behavior as a methodological limitation.
Overall, the article offers a novel contribution by linking body image parameters to differential behavioral responses to UPF intake, but findings require replication in larger, adequately powered trials with tools sensitive to a broader range of eating disorders.
Author Response
- Comment 1: “Clarify the rationale for including specific psychometric instruments and acknowledge limitations of EAT-26 in obese populations.”
- Author’s response 1: We thank the reviewer for this important observation. The selection of psychometric instruments was guided by their relevance to the constructs under investigation, prior use in similar contexts and cultural adaptation and validation for the Brazilian population, as described in the methods section under each variable (lines 139-145; 156-160; 171-172; 178-181). The final set included the modified Yale Food Addiction Scale 2.0 (mYFAS 2.0), which assesses symptoms of food addiction based on Diagnostic and Statistical Manual of Mental Disorders, Fifth Edition (DSM-5) criteria; the Silhouette Rating Scale, developed by Kakeshita et al. [38] and validated for the Brazilian context to assess body image perception and satisfaction; the GAD-7 scale, a widely used and reliable screening tool for generalized anxiety disorder; and the Eating Attitudes Test (EAT-26) to evaluate eating behaviours and attitudes potentially associated with eating disorders. The Silhouette Rating Scale also uses standardized images that minimize subjective bias, as we pointed out in the final section of the discussion (lines 368-373). Additionally, we acknowledge the specific limitations of the EAT-26, initially included for exploratory purposes, as a screening tool rather than a diagnostic measure. Furthermore, it captures risk across a broad range of eating disorders, including anorexia and bulimia, conditions that are less prevalent among individuals with obesity. Nevertheless, these limitations were clearly acknowledged throughout the discussion (lines 345-348).
- Comment 2: “Expand on the theoretical basis for the proposed mechanisms linking body image with eating rate.”
- Author’s response 2: We appreciate the reviewer’s interest in further exploring the mechanisms linking body image and eating rate. While our findings offer preliminary insight into this association, it is important to emphasize that this is a secondary analysis of a randomized controlled trial not originally designed to test mechanistic hypotheses in this domain. Our goal is not to propose definitive explanations, but rather to generate new hypotheses and identify relevant psychological variables that need deeper investigation. Nevertheless, we have contextualized the observed association with theoretical references grounded in existing literature on body image. In the discussion section, we suppose that the tendency to eat slowly among individuals who expressed greater body image dissatisfaction in the UPF group may suggest a form of self-regulation or internal conflict when engaging with a stigmatized food category, potentially driven by concerns about appearance. We believe this careful framing enables the findings to retain relevance without overextending their interpretive scope, as these findings warrant further research.
- Comment 3: “Provide power calculations adjusted for the interaction models used.”
- Author’s response 3: We appreciate this comment. We would like to argue that we prefer not to conduct a post-hoc power analysis in this study, given that we have already acknowledged it is a secondary study and, of course, it is exploratory by nature. The a priori sample size calculation is not tailored to test the present hypothesis. We are afraid that post-hoc power analysis, in the case of a completed study, won't give any additional information, since that the power achieved is only a monotone function of the p-value already reported in the paper (Dziak JJ et al., The Interpretation of Statistical Power after the Data have been Gathered. Curr Psychol. 2020;39(3):870-877. doi: 10.1007/s12144-018-0018-1.) Hence, we would like to follow several guides that suggest avoiding the reporting of achieved power, as it is practically misleading and logically flawed (Heckman MG et al.. Post Hoc Power Calculations: An Inappropriate Method for Interpreting the Findings of a Research Study. J Rheumatol. 2022;49(8):867-870. doi: 10.3899/jrheum.211115; Christogiannis C, et al. The self-fulfilling prophecy of post-hoc power calculations. Am J Orthod Dentofacial Orthop. 2022;161(2):315-317. doi: 10.1016/j.ajodo.2021.10.008.)
- Comment 4: “Address the potential effect of video observation on eating behavior as a methodological limitation.”
- Author’s response 4: We thank the reviewer for this important observation. We have addressed this point in the revised manuscript by acknowledging that video recording may have influenced participants’ eating behaviour due to the well-documented observer effect (lines 365-368). Even with video recording, participants were unaware of the specific study outcomes. We also note that video observation remains one of the least intrusive and a valid method available for capturing real-time eating episodes, especially when compared to self-report, which may introduce greater bias. It is particularly challenging to monitor participants via video without their awareness, due to the need for camera positioning that provides a clear view of the face and body to enable accurate analysis.